# Genome-Wide Association and Genomic Prediction of Alfalfa (*Medicago sativa* L.) Biomass Yield Under Drought Stress

**DOI:** 10.3390/ijms26020608

**Published:** 2025-01-13

**Authors:** Cesar A. Medina, Julie Hansen, Jamie Crawford, Donald Viands, Manoj Sapkota, Zhanyou Xu, Michael D. Peel, Long-Xi Yu

**Affiliations:** 1Department of Agronomy and Plant Genetics, University of Minnesota, Saint Paul, MN 55108, USA; medin297@umn.edu (C.A.M.); zhanyou.xu@usda.gov (Z.X.); 2School of Integrative Plant Science, Plant Breeding and Genetics Section, Cornell University, Ithaca, NY 14850, USA; jlh17@cornell.edu (J.H.); jln15@cornell.edu (J.C.); drv3@cornell.edu (D.V.); 3Breeding Insight, Cornell University, Ithaca, NY 14850, USA; ms3743@cornell.edu; 4Plant Science Research Unit, USDA-ARS, St. Paul, MN 55108, USA; 5Forage and Range Research Laboratory, USDA-ARS, Logan, UT 84322, USA; 6Plant Germplasm Introduction and Testing Research Unit, USDA-ARS, Prosser, WA 99350, USA

**Keywords:** GWAS, genomic prediction, alfalfa, yield, drought stress

## Abstract

Developing drought-resistant alfalfa (*Medicago sativa* L.) that maintains high biomass yield is a key breeding goal to enhance productivity in water-limited areas. In this study, 424 alfalfa breeding families were analyzed to identify molecular markers associated with biomass yield under drought stress and to predict high-merit plants. Biomass yield was measured from 18 harvests from 2020 to 2023 in a field trial with deficit irrigation. A total of 131 significant markers were associated with biomass yield, with 80 markers specifically linked to yield under drought stress; among these, 19 markers were associated with multiple harvests. Finally, genomic best linear unbiased prediction (GBLUP) was employed to obtain predictive accuracies (PAs) and genomic estimated breeding values (GEBVs). Removing low-informative SNPs [SNPs with *p*-values > 0.05 from the additive Genome-Wide Association (GWAS) model] for GBLUP increased PA by 47.3%. The high number of markers associated with yield under drought stress and the highest PA (0.9) represent a significant achievement in improving yield under drought stress in alfalfa.

## 1. Introduction

Water is required for plant survival, growth, and production. However, the water demand for agriculture could double by 2050, while the availability of fresh water is predicted to drop by 50% [1]. If true, this would generate a growing demand alongside an increasing reduction in clean water availability, projecting water scarcity by 2040. Drought stress is the major factor limiting crop productivity worldwide, making drought resistance, or production under drought stress, one of the most important and desirable features in modern agricultural crops. Agriculture can ameliorate this by adopting water-smart cropping techniques to maintain or improve crop yield under climate change conditions, such as water scarcity [2]. Another approach is to develop high-yielding crops that use water more efficiently than current varieties.

Drought resistance mechanisms can be classified as molecular, physiological, or morphological aspects, all of which are interconnected as effects or responses. The molecular mechanisms associated with drought stress (i.e., up- and downregulation of drought stress related genes) include abscisic acid signaling pathway, MAPK signal transduction pathway, cell wall biogenesis, ABC transporters, and ion transport and homeostasis pathways [3,4,5]. These pathways produce physiological changes involving (i) the closure of stomata to decrease water loss via transpiration, (ii) the synthesis of protective proteins like dehydrins and osmoprotectants, and (iii) the activation of antioxidant defense systems to compensate for oxidative stress [6]. The morphological mechanisms include (i) drought escape, when the plant accelerates the reproductive phase (early flowers) or undergoes a period of dormancy while water is limited; (ii) drought avoidance, when the plant maintains a relatively high tissue water potential through stomatal closure to prevent tissue damage; and (iii) tolerance, when plants maintain turgor and continue growth, even at low water potential, mainly by osmolyte adjustment [5]. Understanding the mechanisms involved in the drought stress response is fundamental for developing new alfalfa varieties that are tolerant to drought stress.

Alfalfa (*Medicago sativa* L.) is one of the major forage crops in dairy and livestock production and is often referred to as “the queen of forages”. Although alfalfa has good water-use efficiency under drought stress [7], significant yield losses occur when water deficits become severe [8]. There is a need to develop new cultivars that maintain high biomass yield under drought conditions. Increasing biomass yield in alfalfa is more challenging compared to annual grain crops due to four factors: (i) The perennial growth habit of alfalfa requires the allocation of energy and nutrient storage for winter survival, which can affect yield in the following year. (ii) Unlike annual crops, the perennial nature of alfalfa also necessitates multiple years per breeding cycle. (iii) Increasing biomass yield in alfalfa involves enhancing assimilatory processes (photosynthesis), which is more difficult to achieve than modifying regulatory processes, such as seed retention or redirecting energy to specific organs in grain crops. (iv) The breeding efforts in alfalfa have predominantly focused on pest and disease resistance. However, these resistances do not necessarily increase yield when the crop is grown in the absence of pests [9].

Recently, molecular breeding approaches such as marker-assisted selection (MAS) have been implemented in breeding programs. Molecular approaches like genome-wide association studies (GWASs) [10,11,12] and transcriptomic studies [4,13] have identified key DNA markers, genes, or gene networks associated with drought stress. The closely linked markers identified by GWAS can be utilized through MAS to accelerate the development of improved cultivars. However, many important traits such as yield or tolerance to abiotic stress are quantitative and controlled by many genes, making it difficult to identify a major genetic locus that explains the variation among genotypes. To address this, Meuwissen (2001) proposed a tool that utilizes all markers simultaneously as random effects to predict genetic performance (genomic prediction, GP) [14]. Unlike MAS, GP does not pinpoint specific marker effects; instead, it takes the whole genome into account and trains for a model that can be used to predict the genetic performance of individuals or the genomic estimated breeding value (GEBV). The implementation of GP models in breeding programs is called genomic selection (GS), and its value depends on the predictive ability (PA), which is the correlation between the true value and the GEBV after cross-validation testing.

In perennial crops with long selection cycles, such as alfalfa, the implementation of GS in breeding programs will reduce the time required for a selection cycle, accelerating breeding process. However, the PA of GP models have been found to be low (<0.6) for quantitative traits in alfalfa [15,16,17]. Recent studies have demonstrated that incorporating GWAS marker information into the GP model increases the PA of the model [18,19]. In this work, biomass yield data were collected in a field trial with an imposed deficit irrigation gradient across 18 harvests from May 2020 to August 2023 in a population of 424 half-sib families (HSFs). The objectives of this work were to (i) identify HSFs with high yield under drought conditions, (ii) identify molecular markers and candidate genes associated with high biomass yield under drought stress, and (iii) obtain highly accurate GEBVs of drought-tolerant HSFs to select the resistance families for advanced selection cycles.

## 2. Results

### 2.1. Impact of Drought Stress in Biomass Yield of Alfalfa

Fresh biomass yield was measured in grams per plant from May to September for four years in either four or five cuttings. The number of missing plots was low (≤2.14%) throughout the experiment (Appendix A). The highest yield was at the first harvest (May) each year and decreased as the season progressed and drought stress was applied, especially during 2020 and 2023. The lowest yield was in September 2021 (mean = 123 g × plant^−1^), and the highest was in May 2023 (mean = 401 g × plant^−1^) (Figure 1a).

BLUE values from the first-stage analysis were used to predict the yield by month, year, or overall responses within HSFs using a stagewise approach (ST2). The overall predicted yield was significantly different between months except June and July with an overall yield reduction of 42% between the first and last harvests (Figure 1a). The overall predicted yield by year was significantly lower (*p*-value < 0.05) than observed in 2021 and 2023, but there was no difference for those between 2020 and 2022 (Appendix A). Alfalfa yields naturally decline over time with the overall yield between the first and last years (2020 vs. 2023) decreasing by 7% on average, with a decrease by more than 10% in 86 out of the 436 HSFs.

Pearson’s correlation analysis revealed an autoregressive correlation between yields by month for the same year. The highest correlation was found between June 2021 and July 2021 (*R* = 0.71), and the lowest correlation was between July 2023 and September 2020 (*R* = 0.25). On average, the yield from August 2023 was the least correlated ($\bar{R}$ = 0.37), while that from May 2021 was moderately correlated ($\bar{R}$ = 0.52) (Figure 1b). Pearson’s correlation of the yield averaged by month or by year (ST2) was highest ($\bar{R}$ = 0.89), with the highest correlation found between June and July (*R* = 0.91) and the lowest correlation between May and September (*R* = 0.76) and between the harvests of 2022 and 2023 (*R* = 0.73) (Figure 1c).

The residual variance was greater than the genetic variance (V_G_) at all harvests, representing 82% of the total variance. May 2021 had the highest V_G_ at 31%, while August 2023 had the lowest value of V_G_ (6%) (Appendix A). The mean value of broad-sense heritability (H^2^) was 0.37, with the highest value in May 2021 (0.54) and the lowest in August 2023 (0.16) (Appendix A), while the coefficient of variation was the lowest in May 2021 (14.9%) and highest in August 2023 (33.1%) (Appendix A). There was no significant difference in H^2^ or coefficient of variation (CV) when comparing harvests from the same month or year, but as expected, the highest H^2^ values were related to the lowest CVs.

To evaluate the genotype-by-environment interaction (G×E) between HSFs and the environmental conditions (i.e., normal irrigation and drought stress), the top 10% of high-yield HSFs were compared under severe drought stress conditions (September 2020, 2021, 2022, and August 2023). Seven HSFs (16, 274, 514, 515, 548, 558, and 584) consistently exhibited high yields in all four harvests and ten HSFs (18, 76, 186, 195, 271, 275, 423, 481, 482, and 603) showed high yields in three out of four harvests. The seven top-performing HSFs revealed a yield reduction of 12-61% compared to the control grown under non-stress conditions (Appendix A). For example, HSF 584 has a mean reduction of 46%, while HSF 514 has a mean reduction of 31%.

Additionally, G × E was evaluated by Shukla’s stability variance ($\sigma_{i}^{2}$) index. $\sigma_{i}^{2}$ values ranged from 1383 (HSF 513) to 25,045 (HSF 24). Consequently, the yield of HSF 513 (low G × E) varied by only 120 g in all harvests, whereas HSF 24 (high G × E) had a yield range of 648 g across all harvests (Figure 2d). Some HSFs were stable across environments (low $\sigma_{i}^{2}$) and produced high biomass yields (*y*, in g × plant^−1^), such as 423 ($\sigma_{i}^{2}$ = 8794 and *y* = 393), 271 ($\sigma_{i}^{2}$ = 9281 and *y* = 360), and 274 ($\sigma_{i}^{2}$ = 9884 and *y* = 353) (Appendix A). $\sigma_{i}^{2}$ was lower in ST2 compared with ST1, ranging from 326 (HSF 513) to 8175 (HSF 493) (Appendix A). Again, HSF 423 ($\sigma_{i}^{2}$ = 3888 and *y* = 392), 271 ($\sigma_{i}^{2}$ = 1853 and *y* = 365), and 274 ($\sigma_{i}^{2}$ = 3343 and *y* = 347) have a low $\sigma_{i}^{2}$ and high *y* (Appendix A). HSF 584, despite achieving high overall yields (418 and 422 g × plant^−1^ in ST1 and ST2, respectively), had a low stability (7353 and 16,886) across the different harvests, indicating a high G × E (Figure 2d).

### 2.2. Marker Distribution Largely in Coding Regions

In total, 1,950,604,130 sequence reads, 100 nucleotides in length, were obtained from FASTQ files, with a mean read quality Phred score of 35.57. After preprocessing the raw FASTQ files, 99.8% of the reads were retained (1,947,903,302 reads). The overall alignment rate was 82% to the *Medicago sativa* cultivar XinJiangDaYe genome (monoploid version). After filtering, 51,081 high-quality SNP markers were obtained and plotted according to their chromosome using a 1 Mb window size (Figure 2a). The distributions of markers by chromosomes were as follows: chromosome 1 (Chr. 1) = 7875 markers, Chr. 2 = 5783 markers, Chr. 3 = 7259 markers, Chr. 4 = 7448 markers, Chr. 5 = 6493 markers, Chr. 6 = 3379 markers, Chr. 7 = 6407 markers, and Chr. 8 = 6437 markers. Chromosome six had the lowest number of markers, with a density of 42 SNPs/Mb, and a maximum gap between two markers of 1.8 Mb, and chromosome one had the greatest number of markers, with a density of 95 SNPs/Mb, and a maximum gap between two markers of 0.8 Mb. Although there was a high number of markers by chromosome (>30 k SNPs), the genomic coverage across the chromosomes was not uniform. The genomic coverage associated with the subsets clearly showed a decreasing trend in centromeric regions on chromosomes five and six (Figure 2a). The mean depth of sequencing by individuals showed that only two samples had coverage lower than 16× (Figure 2b). The mean depth of sequencing by sample was 24.2 reads, with a minimum and maximum coverage of 10× and 51.3×, respectively. All markers obtained had high sequencing depth, which is sufficient to identify allele dosages with high certainty in tetraploid organisms. The linkage disequilibrium decay at *r*^2^ = 0.1 in all the accessions was 7.5 Mb (Figure 2c). Marker annotation identified that 15,877 (31.1%) SNPs were in non-coding regions, while 35,204 (68.9%) were found in coding regions. Among those SNPs in coding regions, 13,733 (26.9%) were in intronic regions and 21,470 (42%) were in coding regions including synonymous or missense variants (Figure 2d).

Principal component analysis (PCA) was conducted on 424 HSFs with genotypic information to determine the genetic relationships among families. The first and second components collectively accounted for 3.69% of the total genetic variance, with 2.4% attributed to the first component and 1.29% attributed to the second component (Appendix A). The samples were clustered using K-means into three clusters by PCA based on the backcross pedigree information (Appendix A). Discriminant analysis of principal components (DAPC) was successfully used for assigning HSFs to specific clusters (Appendix A). Approximately 500 principal components were included in the preliminary data transformation step, enabling DAPC to explain 80% of the total genetic variation. Cluster one was the most diverse, comprising 285 HSFs from 27 parents. Cluster two comprised 71 HSFs, where 31 HSFs came from drought-susceptible parent 15. Cluster three comprised 68 families, with 37 HSFs coming from drought-susceptible parents 16 (9 out of 12) and 18 (28 out of 29) (Appendix A).

### 2.3. Association Studies Identify Low Redundancy of Markers Associated with Yield Under Drought Stress

A GWAS for biomass yield was conducted for each harvest (ST1), grouped by month, year, or overall yield (ST2), or comparing the first (non-stress) and last (severe drought stress) harvests by year (DS). A total of 137 unique markers were identified, with −log_10_(*p*-values) up to 11.44 (Appendix A), showing significant peaks in June 2022 (chromosome 2), July 2022 (chromosome 3), and July 2023 (chromosome 6) (Appendix A). Marker counts varied by harvest; no significant markers were found for July 2020 and August 2021, while September 2021 and May 2023 had the most markers (48 and 41, respectively). Of the 137 markers, 123 were associated with ST1, 25 with ST2, and 80 with DS across different years. Notably, three regions of approximately 10 Mb on chromosomes 2, 3, and 7 were linked to DS across 2021–2023 (Figure 3a). Shared markers included one between ST1 and ST2, 48 between ST1 and DS, and 19 among all three analyses (Figure 3b).

Chromosome 7 had the highest number of markers (30), and chromosome 1 had the fewest (7). High marker densities were found in regions on chromosomes 2, 3, and 7. On chromosome 2, eleven markers clustered in a 6.9 Mb region (40–46.9 Mb); on chromosome 3, seven markers were found in a 4.7 Mb region (55.7–60.4 Mb); and on chromosome 7, there were two marker clusters: one with six markers in a 3.4 Mb region (19.2–22.7 Mb) and another with nine markers in a 5.3 Mb region (11.2–16.7 Mb) (Figure 3c). Eighty-five markers (62%) were in coding regions, with six near coding regions (<1 kb) (Figure 3d).

Gene annotation of significant marker positions revealed 86 candidate genes, including six transcription factors (*Overexpressor of cationic peroxidase* 3, *C2H2 transcription factor, transcription factor TCP15* [*Teosinte branched1/Cycloidea/Proliferating cell factor*], *Transcription factor TGA7*, *Zinc finger protein ZF3*, and *Auxin response factor 9*), three pentatricopeptide repeat-containing proteins, and four protein kinases (*Proline-rich receptor-like protein kinase* 13, *Cyclin-dependent kinase D-3-like*, *Serine/threonine protein kinase D6PKL2*, *Calcium-dependent protein kinase* 13, and *Mitogen-activated protein kinase* 2) (Appendix A). These 137 SNPs were further used to prioritize candidate genes for drought stress via cageminer [20].

Transcriptomic data from a previous study [13] helped build a gene coexpression network using the BioNERO v1.12.0 R package [21]. Forty-six SNPs identified 50 high-confidence candidate genes for drought stress. Nine genes were prioritized by multiple markers (<1 Mb), such as an *F-box factor* located 94 and 102 kb downstream of three markers on chromosome 3 (chr3.1_6556553, chr3.1_6563931, and chr3.1_6563992).

Additionally, 12 SNPs were prioritized by more than one candidate gene; for example, the marker chr5.1_1429434 prioritized three genes (*G protein-coupled receptor*, *Vicinal oxygen chelate glyoxalase* and *Poly(A)-Binding Site domain*). Gene prioritization identified key genes, including two transcription factors (*MYB102 transcription factor* and *Squamosa promoter binding protein*), four cold-regulated genes (*Late embryogenesis abundant* II, *Low-temperature-induced 65 kDa*, *Glycine-rich RNA-binding protein* 2, and *Cold-regulated protein*), and nine drought-regulated genes (*MYB102 transcription factor*, *Indole-3-acetic acid-induced, Late embryogenesis abundant* II, *Low-temperature-induced 65 kDa*, *Glycine-rich RNA-binding protein 2*, *Abscisic acid receptor PYL4*, *4-hydroxyphenylpyruvate dioxygenase*, *Aquaporin,* and *Pyrroline-5-carboxylate reductase*) (Appendix A).

### 2.4. Genomic Prediction Increases the Accuracy When Non-Informative Markers Are Removed

The genomic best linear unbiased prediction (GBLUP) method was used to measure relationships between individuals using a marker-based genomic relationship matrix (GRM). The weighted GBLUP (WGBLUP) method incorporated SNP weights into the G matrix, with SNP weights derived from −log_10_(*p*-values) of the GWAS additive model for SNPs associated with overall yield. Notably, 48,335 markers (94.6%) were not significant (*p*-values > 0.05 or −log_10_(*p*-values) < 1.3), leaving 2746 significant markers. However, the percentages of coding and non-coding markers were similar to those in the total marker set (Figure 4a).

The four G matrices showed high correlations (0.91 to 0.99) but significantly differed in terms of their diagonal values (*p*-value < 0.05). The diagonal elements represent 1 plus the realized inbreeding level for the corresponding individuals. Values of 1 indicate non-inbred individuals, values < 1 indicate an excess of heterozygotes, and values > 1 indicate an excess of homozygotes. The highest diagonal value was observed in G3 (2.55), followed by G4 (1.44), G1 (1), and G2 (0.64) (Figure 4b). Off-diagonal values were close to zero, ranging from −0.0051 (G3) to −0.0013 (G2).

Biomass yield by harvest (ST1) and averaged by month, year, and overall (ST2) was used to calculate genomic estimated breeding values (GEBVs) and predictive accuracy (PA) for each G matrix. G1 yielded the lowest PA (mean = 0.1), while G3 had the highest PA (mean = 0.55) in ST1 (Figure 4c). For instance, the PA of G3 in May 2021 was 3.75 times higher than G1 (Appendix A). No significant difference was found between G3 and G4, though G3 showed a 6% increase in PA over G2. In ST2, the PA was higher than in ST1, with G1 producing the lowest PA (0.16 for 2022) and G3 producing the highest (0.9 for overall yield) (Figure 4d).

## 3. Discussion

In this study, we evaluated alfalfa biomass from 424 half-sib families (HSFs) from 18 harvests over four years under water-stressed and non-stressed conditions. The high number of harvests with varying irrigation deficits provided a complex dataset for modeling spatial and temporal variations. A spatial mixed linear model using splines was employed to correct spatial variation by harvest, providing precise genetic estimates [22,23], which were then used in a second-stage modeling to predict yields by HSF by month, year, and overall. This approach has been useful for identifying molecular markers associated with fall dormancy and forage quality in alfalfa [24].

As expected, biomass yield between harvests showed high temporal correlations, particularly between May and June. In contrast, the last harvest (August 2023) exhibited the lowest correlation ($\bar{R}$ = 0.33) and the lowest heritability (0.16), likely due to severe drought stress. Spatial and temporal correlation is expected in perennial crops [25]. Therefore, Pearson correlation values were based on genetic estimates that included either spatial effects (Figure 1b) or both spatial and temporal effects (Figure 1c). The high correlations observed during the 2020 and 2021 harvests, especially between the June and July 2021 harvests (*R* = 0.71), were attributed to similar environmental conditions. Conversely, the lowest correlation observed between July 2023 and September 2020 (*R* = 0.25) resulted from a difference of 12 harvests, spanning different years and seasons. The overall yield decreased significantly over the year, with May yielding the highest biomass and September the lowest. These trends were consistent with previous studies showing that drought stress reduces biomass over time [10].

It has been reported that in alfalfa, the first harvest of the year usually produces the highest biomass, and then decreases as the season progresses, as the increase in drought stress reduces the yield biomass [26]. However, in our study, annual yield variability was determined by weather conditions. The highest biomass yields occurred in 2020 and 2022, while 2021 and 2023 showed significant reductions, indicating environmental impacts rather than a steady decline in biomass. Notably, only 2.14% of the plots were missing, suggesting plants were not stressed sufficiently to cause mortality. Genotypic analysis revealed that HSFs 515 and 584 had high yields but were unstable across harvests, while HSFs 423, 271, and 274 showed high yields and stability, with HSFs 271 and 274 originating from the same F1 backcross (1622-26 × 1613-26), while HSFs 423, 515, and 584 are from different backcrosses.

Several studies have identified genetic loci associated with drought stress in alfalfa. For instance, Ray et al. (2015) identified 25 QTLs associated with drought-induced biomass yield [27], and Zhang et al. (2015) reported 19 markers for drought resistance in 198 accessions [12]. In this work, we identified 137 markers associated with biomass yield under drought stress with the highest PVE of 8.1% (chr7.1_67995967). Eighty-six (63%) markers were located in coding regions of annotated genes with multiple functions. Many markers were found in genes with known functions related to stress response, such as pentatricopeptide repeat-containing proteins (PPRs) and transcription factors (TFs), which are involved in abiotic stress tolerance (Appendix A).

To further refine candidate gene selection, we used the cageminer R package, which ranks genes based on differential expression near significant SNPs [20]. Here, we used the previous transcriptomic information from Wilson (drought-tolerant) and Saranac (drought-sensitive) alfalfa varieties under drought stress [13] highlighting genes involved in drought stress, including two TFs (*MYB102* and *SBP*) and four cold-regulated genes (*LTI65*, *LEA II*, *GRP2*, and *COR*), and ten genes involved in drought stress (*ARG2*, *EXO70*, *LTI65*, *LEA II*, *GRP2*, *MYB102*, *PYL4*, *HPPD*, *TIP1:1*, and *PSCR*) (Appendix A).

The use of genomic selection for complex traits such as biomass yield in alfalfa is potentially more useful than marker-assisted selection, where multiple markers with low phenotypic variance are explained. Previous studies have tested the influence of the number of markers on the PA [17,28]. We demonstrated that the weighted GBLUP method (WGBLUP) increased PA by 80% in alfalfa yield under salt stress using a vector weight of −log_10_(*p*-values) from the GWAS model [19]. In this study, we found that filtering SNPs with *p*-values > 0.05 from the GWAS model resulted in a 3.75-fold increase in predicted ability compared to the regular GBLUP model, reaching up to 90% PA in overall yield. A similar approach was applied by Zhang et al. (2023), where using the top 3000 GWAS markers (out of 875,023) achieved a PA of 64% for FD in alfalfa [18].

Although we trained our GP models 10 times with 10-fold cross-validation (90% training and 10% testing), there is a risk of overfitting, as the models focus on markers relevant to the current population. To address overfitting, testing the marker subset in a new alfalfa population could be beneficial. However, due to the randomness of GBS-generated markers, this is challenging. Recently, a 3 k panel of DArTag markers for alfalfa was generated, and the reproducibility of this panel can facilitate the accomplishment of this objective [29]. These findings highlight the potential of genomic selection for complex traits like biomass yield in alfalfa.

## 4. Materials and Methods

### 4.1. Plant Materials and Field Trials

A backcross population comprising 436 alfalfa half-sib families (HSFs), each containing at least 18 individuals, was developed for testing. In summary, 30 drought-susceptible parental plants from Guardsman II were selected and crossed with 32 drought-resistant parents from alfalfa breeding programs in Utah and Washington to generate an F1 generation (1622). Seeds from the F1 generation were used to produce plants which were then backcrossed with the susceptible parents in a greenhouse located at Cornell University (Ithaca, NY, USA), resulting in a backcross population containing 436 HSFs.

Seeds from HSF were germinated in the greenhouse, and three-month-old plants were selected for transplanting to the field. A total of 7848 individual plants were established in the field trial, using a randomized complete block design with three replicates, each containing six plants per plot. The field trial was spatially arranged in 23 columns and 57 rows, with 1308 plots. Within each plot, six plants were grown in two rows, with 30 cm between individual plants and 90 cm between plots. Border rows were planted with the same spacing to eliminate any border effects.

The study was established in 2019 at the Roza experimental field in Prosser, WA (46°17′23.6″ N and 119°43′33.5″ W). The average precipitation at the site was 14.6 mm and the average temperature was 11.4 °C (Appendix A) [30]. Prior to planting, the soil was prepared using conventional tillage, and a drip irrigation system was installed. During the establishment phase in 2019, the trial was fully irrigated, i.e., the field was irrigated once per month for 24 h at a rate of 0.26 gallons per minute through hoses with drip holes spaced every 18 inches. From 2020 to 2023, irrigation was applied once per month as described above from April to June and restricted from July to September to induce drought stress (Appendix A).

The aboveground fresh weight biomass (biomass yield) was collected from May to September each year from 2020 to 2023, in a total of 18 harvests (Appendix A). Harvests were conducted at the early bud stage using a 36A RCI forage harvester (RCI Engineering, Mayville, WI, USA), which recorded biomass yield by plot. The fresh biomass weight was then transformed to grams × plant^−1^.

### 4.2. Genotypic Modeling

Genotypic modeling was performed using a spatial mixed linear model (MLM) with the SpATS package v.1.0-18 [31] in a single-stage approach (i.e., by harvest). The MLM corrects phenotypic responses with two-dimensional penalized splines to obtain best linear unbiased estimates (BLUEs) and is defined as(1)$$y=X\beta +f\left(r,c\right)+Z_{u}u+Z_{g}g+\varepsilon$$
where the vector $y=\left(y_{1},\ldots ,y_{1308}\right)$ represents the phenotypic response of 1308 plots, X is the association design matrix, and β is a vector of fixed effects. The function $f\left(r,c\right)$ represents a semiparametric bivariate function of the rows $r=\left(r_{1},\ldots ,r_{57}\right)$ and columns $c=\left(c_{1},\ldots ,c_{23}\right)$, corresponding to the vector of random spatial effects. u is a vector of random row and column effects, accounting for discontinuous field variation, and $Z_{u}$ is the associated matrix. g is the genotypic vector of 436 HSFs, with $Z_{g}$ as the associated design matrix treated as a fixed effect. ε is the random error vector $\left(\varepsilon_{1},\ldots ,\varepsilon_{1308}\right)$, which is distributed as $\varepsilon \;\sim N\left(0,\sigma_{\varepsilon }^{2}I_{1308}\right)$. Broad-sense heritability (H^2^) was calculated using the function getHeritability from the SpATS R package v.1.0-18 [31].

For multi-environmental trials (METs), BLUE yields were used as input for stagewise analysis to estimate the best linear unbiased predictor (BLUP) values. BLUPs were computed using a factor analytic covariance structure for METs with ASReml v4.2 software [32]. Stagewise BLUP values of yield were predicted to estimate genotype-by-month, genotype-by-year, and overall yield by genotype interactions.

### 4.3. Yield Stability

Genotype stability variance was measured using the Shukla (1972) method, which assumes that a genotype is stable if its response to the environment is parallel to the mean response of all genotypes in the trial. Therefore, genotypes with lower values are considered more stable across environments [33]. Shukla’s stability variance ($\sigma_{i}^{2}$) was calculated using the ASRtriala v1.0.1 R package [34].

### 4.4. DNA Extraction and Sequencing

Genomic representations of 436 individual plants (one plant per family) were generated by collecting two to three leaflets (~100 mg) per genotype for DNA extraction and genotyping. Genomic DNA was extracted using a Qiagen DNEasy 96 Plant Kit (Qiagen, Valencia, CA, USA) following the manufacturer’s instructions. DNA concentration and quality were measured using a NanoDrop ND1000 spectrophotometer (NanoDrop Technologies, Inc., Wilmington, DE, USA). The extracted DNA was sequenced at the University of Minnesota Genomics Center (Minneapolis, MN, USA) for genotyping-by-sequencing, following the protocol described by Elshire et al. (2011) [35]. Sequencing was performed on an Illumina NovaSeq 6000 platform (San Diego, CA, USA), producing single-end reads of 100 bp each. A total of 1,950,604,130 reads were obtained from the population.

### 4.5. Genotyping by Sequencing and SNP Calling

FASTQ files were cleaned to remove PCR adapters and low-quality bases at both ends using fastp v0.23.4 [36]. Cleaned files were aligned to the *Medicago sativa* cultivar XinJiangDaYe monoploid genome [37] using NGSEP v4.2.0 [38], with the ReadsAligner function used to generate BAM files. These files were sorted with Picard tools v3.1.1 [39]. Variants were called using NGSEP’s MultisampleVariantDetector with PCR duplicate control by setting the maxAlnsPerStartPos parameter to 100 for higher sensitivity, producing a VCF file. The VCF was filtered using NGSEP’s VCFFilter based on the following parameters: (i) base quality > 30 Phred score; (ii) minimum allele frequency of 0.05; (iii) genotyped positions in at least 70% of samples; (iv) minimum genotyping quality of 40; (v) ploidy = 4; and (vi) imputation via hidden Markov models. Redundant markers were removed with the snp.pruning function in ASRgenomics v1.1.4 [40].

After filtering, 51,082 high-quality markers were obtained and converted into the GWASpoly format [41] using NGSEP’s VCFConverter function.

### 4.6. Population Structure Analysis

Population structure was analyzed using principal component analysis (PCA) and discriminant analysis of principal components (DAPC) with the adegenet R package v.2.1. 10 [42]. DAPC was applied to define clusters (k) of related HSFs without prior information, using multiple principal components for initial data transformation. The find.clusters function was used to predefine groups with the following parameters: method = Ward’s, criterion = goodfit, stat = Bayesian Information Criterion, and k = 3. The dapc function then used these predefined group memberships, retaining 80% of the variance and 3 axes in the discriminant analysis.

### 4.7. Association Mapping

The association mapping analysis was performed using the R package GWASpoly with a Q + K linear mixed model as follows [41]:(2)y=Xβ+ZSτ+ZQv+Zu+ε
where y corresponds to a vector of observed phenotypes; β is a vector of fixed effects; X is an incidence matrix used to model environmental effects; v is the subpopulation vector effect; Q is an incidence matrix for a population of size m; u is a polygenic effect vector; Z is a matrix of incidence mapping genotypes to observations; τ is an SNP effect vector; S is a structure incidence matrix; and ε is a residual vector [41]. Markers associated with yield were identified using a threshold of Bonferroni >0.05, and they were annotated using the alfalfa pan-transcriptome information [13].

Genes in the same location of the SNP marker were annotated using the UniProt database [43]. Additionally, candidate genes associated with drought stress were prioritized by the cageminer v1.10.0 R package [20] using the transcriptomic information of alfalfa under drought stress [13].

### 4.8. Genomic Prediction

Genotypic responses of biomass yield modeled by harvest (ST1), by month, by year, or overall yield (ST2) were used to estimate genomic estimated breeding values (GEBVs) and predict breeding ability using genomic prediction models based on genomic best linear unbiased prediction (GBLUP). GBLUP uses molecular markers to construct the genomic relationship matrix (G matrix) [44]. The weighted GBLUP (WGBLUP) method incorporates unequal marker weights into the G matrix, which is defined as(3)$$WG=\frac{{ZDZ}^{\prime }}{2\Sigma p_{i}\left(1-p_{i}\right)}$$
where Z is an identity matrix for the markers and *p_i_* is the observed minimum allele frequency, and D is a diagonal matrix where each element of the diagonal corresponds to SNP weights derived from log_10_(*p*-values) of the GWAS additive model for overall yield.

Genomic prediction (GP) uses all markers, rather than selecting specific trait-associated markers, to predict GEBV (Appendix A). Four genomic relationship matrices were generated: G1 used all 51,081 SNPs (Appendix A), G2 included SNP weights (Appendix A), G3 retained only SNPs with *p*-values < 0.05, resulting in 2746 SNPs (Appendix A), and G4 included both the 2746 SNPs and their weights (Appendix A).

The GP models were trained 10 times using 10-fold cross-validation with 90% of the genotypes used for training and 10% for testing, using the ASReml v4.2 software [32]. Predictive ability was calculated as the Pearson correlation coefficient between GEBVs and the phenotypic values of the test population.

## 5. Conclusions

To conclude, we evaluated the changes in biomass yield in response to drought stress. (i) We identified HSFs with high yield under drought conditions, (ii) we identified molecular markers and candidate genes associated with drought stress, and (iii) highly accurate genomic estimated breeding values were obtained and can be used for selecting drought-tolerant HSFs. Although biomass yield is a complex trait controlled by multiple genes that interacts with environmental factors in an intricate way, this work reports the broadest list of SNP markers in response to drought stress to date, revealing that two major classes of genes predominate in drought resistance, transcription factors and protein kinases, indicating the complexity of gene regulation and cellular signaling during the drought response. This research effort aimed to uncover the physiological responses of alfalfa plants to drought and used these findings to develop marker-assisted selection methods to accelerate the development of new alfalfa cultivars with high biomass yields under water deficit conditions.

## Figures and Tables

**Figure 1 ijms-26-00608-f001:**
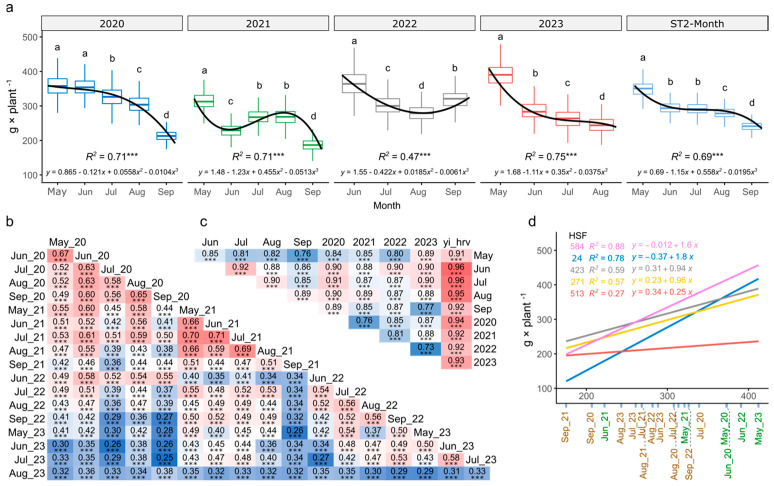
Biomass yield at multiple harvests in alfalfa half-sib families. (**a**) Boxplot and cubic regression of yield by single harvest and overall yield by month (ST2-month), harvests within a year followed by a different letter are significantly different (0.05). *R*^2^ corresponds to the adjusted coefficient of determination of the regression, and asterisks (***) correspond to the *p*-value < 0.001 of the fitted regression. (**b**) Heatmap of the Pearson correlation coefficient of biomass yield by single harvest. (**c**) Heatmap of the Pearson correlation coefficient of biomass yield by month, by year, or overall biomass yield across all harvests (yi_hrv). All correlations were significant (***) at *p*-value < 0.001. The color scale starts from 0 (blue) for low correlation to 1 (red) for high correlation. (**d**) Genotype-by-environment interaction sensitivity of five half-sib families. The *x*-axis represents the mean harvest values, and the *y*-axis represents the predicted biomass yield. *R*^2^ corresponds to the adjusted coefficient of determination for the linear regression. Harvests in green and brown represent normal irrigation and drought stress conditions, respectively.

**Figure 2 ijms-26-00608-f002:**
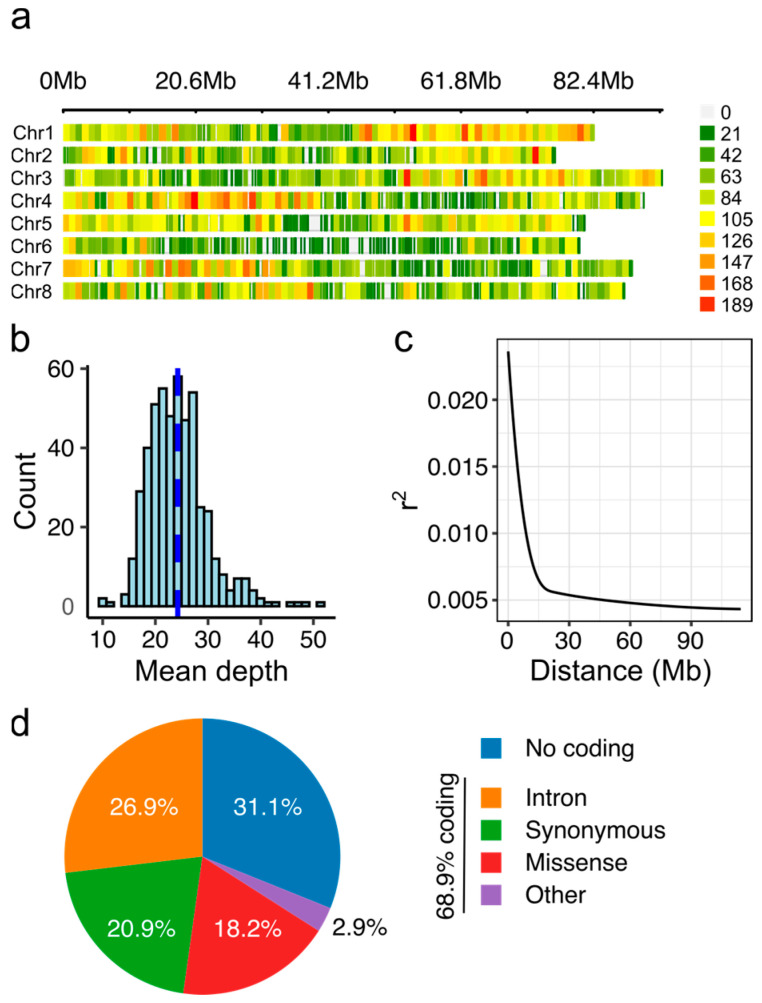
Single-nucleotide polymorphism (SNP) markers identified in alfalfa half-sib families. (**a**) Distribution of SNP markers across eight chromosomes using a 1 Mb window size. Colored lines represent marker density, as shown in the legend on the right. (**b**) Histogram of mean depth reads of the half-sib families. The blue dotted line corresponds to the mean depth. (**c**) Linkage disequilibrium decay of half-sib families. (**d**) Pie chart of marker annotation. Other includes 5′ or 3′ UTR variants, splice variants, and gain or loss in start/stop codon.

**Figure 3 ijms-26-00608-f003:**
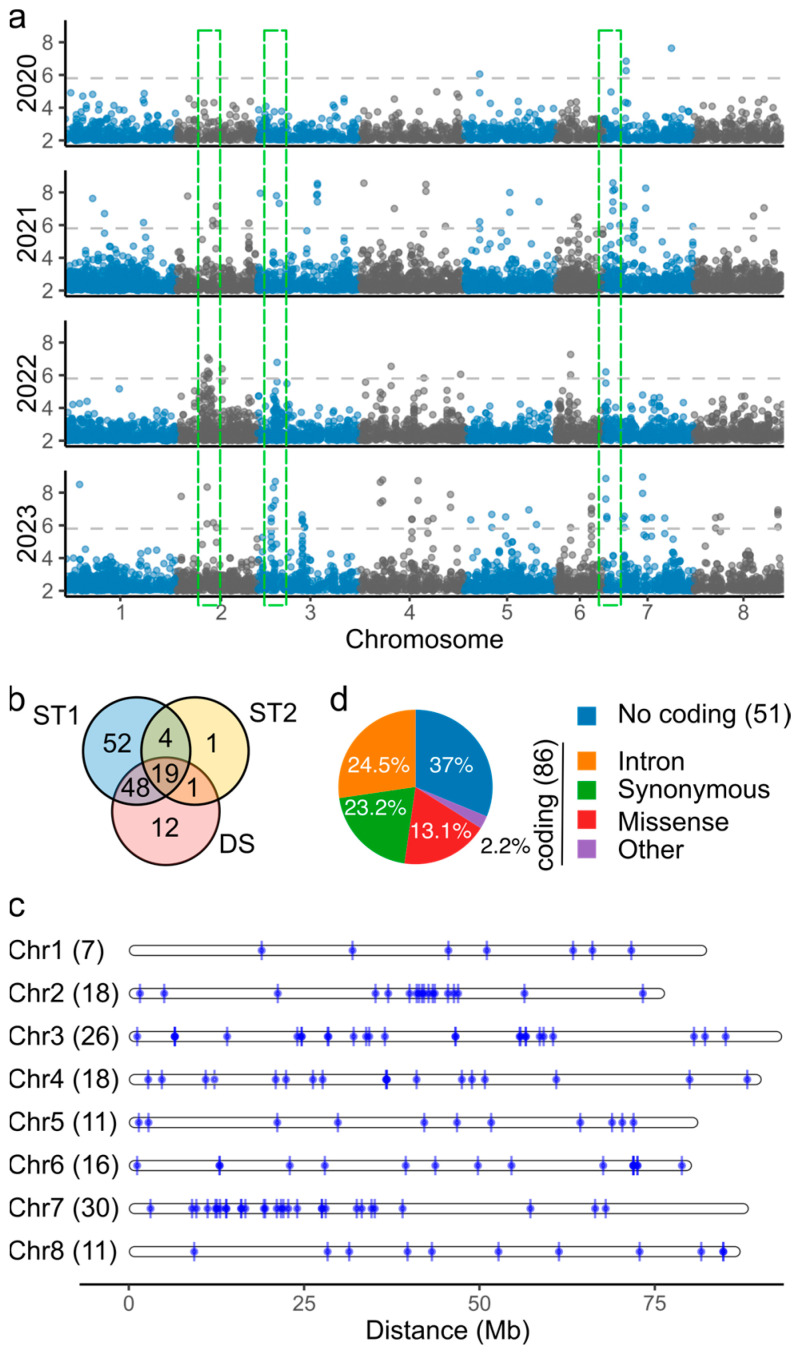
Classification and distribution of significant markers associated with biomass yield. (**a**) Manhattan plots of biomass yield based on drought stress modeling (DS), using first and last harvests of each year as longitudinal data. Green boxes correspond to genomic regions of approximately 10 Mb on chromosomes two, three, and seven, containing shared markers from 2021, 2022, and 2023. (**b**) Venn diagram showing the distribution of associated markers by single-harvest modeling (ST1), month/year modeling (ST2), or drought stress modeling (DS). (**c**) Genomic position of significant markers associated with biomass yield in the alfalfa genome. The *x*-axis indicates the distance in megabases (Mb). The numbers in brackets represent the total number of markers per chromosome. (**d**) Pie chart of marker annotation of 137 significant markers. Numbers in brackets correspond to the number of markers in coding and in non-coding regions. Other includes 5′ or 3′ UTR variants, splice variants, and gain or loss in start/stop codon.

**Figure 4 ijms-26-00608-f004:**
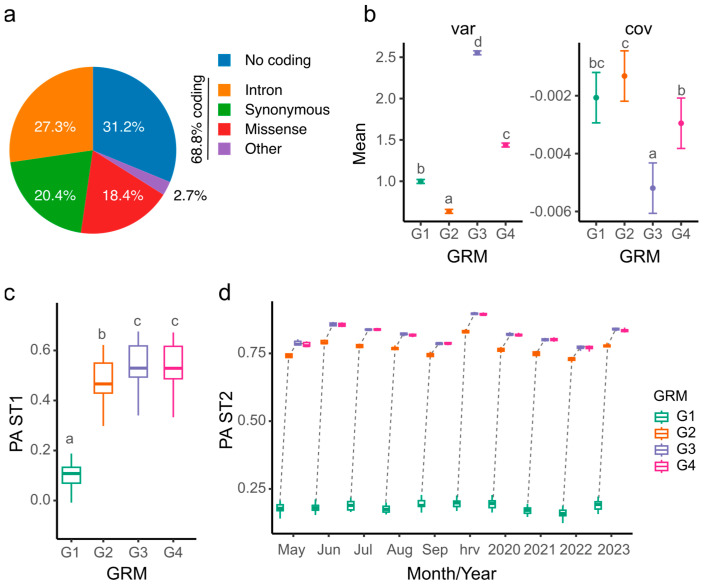
Differences in genomic relationship matrices (GRMs) and comparisons of predicted abilities (PAs). (**a**) Pie chart of marker annotation of 2746 SNPs with a −log_10_(*p*) > 1.3 (i.e., *p*-value < 0.05) from the GWAS additive model. Other includes 5′ or 3′ UTR variants, splice variants, and gain or loss in start/stop codon. (**b**) Line plot of diagonal (var) and off-diagonal (cov) values of four GRMs. Error bars correspond to 95% confidence interval. (**c**) Boxplot of PAs of yield in 18 harvests modeled by a single environment (ST1) in four GRMs. (**d**) Boxplot of PA yield averaged by month, by year (ST2), or by overall yield (hrv) in four GRMs. Dotted lines connect same month/year. GRMs were generated as follows: The G1 matrix was generated using all markers. The G2 matrix was generated using all markers and weights from −log_10_(*p*-values) from the additive model between SNP markers and the overall predicted yield. The G3 matrix was generated using only the 2746 SNPs with a −log_10_(*p*) > 1.3 (i.e., *p*-value < 0.05) from the GWAS additive model. The G4 matrix was generated using the SNPs and weights from markers with a −log_10_(*p*) > 1.3. Different letters indicate significantly different means (*p*-value < 0.05) according to Tukey’s pairwise comparisons.

## Data Availability

Large datasets, including genotypic matrices for GWAS and GP, raw biomass yield, BLUE, BLUP, and GEBVs, are available in figshare (https://doi.org/10.6084/m9.figshare.26254265.v1). The FASTQ files of GBS were deposited in the NCBI Sequence Read Archive under accession number PRJNA1133733, and BioSample accession numbers SAMN42385826-SAMN42386310 (https://dataview.ncbi.nlm.nih.gov/object/PRJNA1133733?reviewer=didm8uahc3u4gdujk7qpshjn88, accessed on 6 December 2024).

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
