# Peer review of "Genome-Wide Association and Genomic Prediction of Alfalfa (Medicago sativa L.) Biomass Yield Under Drought Stress"

_ijms, 2025, doi:10.3390/ijms26020608_

Round 1
Reviewer 1 Report
Comments and Suggestions for Authors
Summary
The manuscript evaluates the yield performance of various experimental populations of alfalfa (Medicago sativa L.) under drought stress conditions. The study integrates a comprehensive genotyping approach followed by a Genome-Wide Association Study (GWAS). While the manuscript is strong overall, minor revisions are needed to improve clarity and completeness.
Title
- Include the full species name (Medicago sativa L.) in the title to provide specificity.
Abstract
- Line 15: Replace the term “family” with a more precise term such as “genotype,” “population,” or “accession,” as “family” could be confused with the botanical family Leguminosae.
- Clarify the experimental factors ∗∗G×E(Genotype×Environment
Introduction
- Expand the introduction to explore gene expression pathways associated with drought stress. Include discussion on the molecular and physiological mechanisms involved. Consider incorporating the following references to enrich this section: https://doi.org/10.1016/j.stress.2024.100657; https://doi.org/10.1016/j.plaphy.2021.05.008
Results
- Genotype Performance:
- Provide details about the genotype, population, or family with the highest yield under drought conditions.
- Indicate how this top-performing genotype diverged from the control grown under non-stress conditions.
- Pearson’s Correlation:
- Discuss the highest and lowest correlations observed across harvest months for better clarity.
- Figures:
- There is a mislabeling issue. The pie chart described in the results appears to correspond to Figure 4a and not Figure 3d. Correct the figure references for consistency.
Materials and Methods
- Drought Stress Environment:
- Elaborate on how the drought stress conditions were established. Were specific irrigation levels reduced? Was there a controlled drought simulation?
- Climatic Data:
- Provide rainfall and temperature data for the experimental period (across the three years of harvest). This will help contextualize the drought conditions.
Author Response
## Reviewer 1
Title
- Include the full species name (Medicago sativa) in the title to provide specificity.
Done
Abstract
- Line 15: Replace the term “family” with a more precise term such as “genotype,” “population,” or “accession,” as “family” could be confused with the botanical family Leguminosae.
Done, changed to breeding families.
- Clarify the experimental factors ∗∗G×E(Genotype×Environment)
The G×E experimental factors were clarified in L130.
Introduction
- Expand the introduction to explore gene expression pathways associated with drought stress. Include discussion on the molecular and physiological mechanisms involved. Consider incorporating the following references to enrich this section: https://doi.org/10.1016/j.stress.2024.100657; https://doi.org/10.1016/j.plaphy.2021.05.008
Thank you for the comment, we incorporate gene expression pathways, the physiological changes and the reference suggested (https://doi.org/10.1016/j.plaphy.2021.05.008) in L39-46.
Results
- Provide details about the genotype, population, or family with the highest yield under drought conditions.
Thank you, we include this information in the L129-134.
- Indicate how this top-performing genotype diverged from the control grown under non-stress conditions.
Thank you, we include this information in the L134-137 and Figure S3.
- Discuss the highest and lowest correlations observed across harvest months for better clarity.
We included a discussion about the highest and lowest correlations in the L311-320.
- There is a mislabeling issue. The pie chart described in the results appears to correspond to Figure 4a and not Figure 3d. Correct the figure references for consistency.
The difference between Figure 3d, and 4a is that in Figure 3d the pie chart of corresponds to marker annotation of 137 significant markers by GWAS and in Figure 4a corresponds to 2,746 markers with a -log10(p) > 1.3 (i.e., p-value < 0.05). Figure 3d is described in L227 and Figure 4a is described in L272.
Materials and Methods
- Elaborate on how the drought stress conditions were established. Were specific irrigation levels reduced? Was there a controlled drought simulation?
In this version of the manuscript, we elaborate it the irrigation system in L388-393. Full irrigation (once per month for 24 hours at a rate of 0.26 gallons per minute through hoses with drip holes spaced every 18 inches) was applied from April to June and restricted from July to September to induce drought stress.
- Provide rainfall and temperature data for the experimental period (across the three years of harvest). This will help contextualize the drought conditions.
Thank you for the comment, the information was included in this version of the manuscript in the L386-387 and in the figure S7.
Reviewer 2 Report
Comments and Suggestions for Authors
In the manuscript “Genome-Wide Association and Genomic Prediction of Alfalfa Biomass Yield under Drought Stress”, the authors tested the effect of drought stress on Alfalfa yield formation, and molecular markers and candidate genes associated with drought stress and genomic estimated breeding values were used to predict Alfalfa yield under drought stress. Overall, the manuscript is interesting, and can help to improve yield-tolerant selection.
1. The keywords should be changed, as the word ‘yield’ is a key word in the manuscript.
2. Line 347: Seeds from the F1 generation were used to…
3. Line 351: were selected for transplanting to the filed.
4. Line 354: comprising of 1,308 plots?
5. The reference part is not very new. I have noticed that the references provided are mainly five years ago. I think that more newly associated references should add.
Author Response
## Reviewer 2
- The keywords should be changed, as the word ‘yield’ is a key word in the manuscript.
Done.
- Line 347: Seeds from the F1 generation were used to…
Done: changed seed to seeds (L375)
- Line 351: were selected for transplanting to the filed.
Done: changed in the field for to the field (L380)
- Line 354: comprising of 1,308 plots?
Done: changed comprising by with (L382)
- The reference part is not very new. I have noticed that the references provided are mainly five years ago. I think that more newly associated references should add.
Thank you for your comment. We have updated the reference section by revising some existing references and adding six new ones.
Reviewer 3 Report
Comments and Suggestions for Authors
The manuscript title “Genome-Wide Association and Genomic Prediction of Alfalfa Biomass Yield under Drought Stress” has scientific worth, as it focuses on one of the important goals of breeding alfalfa, “the development of drought-resistant alfalfa (Medicago sativa L.) that maintains high biomass yield”, also this study will be beneficial for enhancing productivity in water-limited regions. The presentation of this MS is good. This research contributes valuable insights for breeding programs aimed at enhancing drought resistance in alfalfa.
Summary of this study:
This study involved the genotyping and phenotyping of 424 half-sib families of alfalfa to identify molecular markers associated with biomass yield under drought stress and to predict high-merit plants. Biomass yield was measured over 18 harvests from 2020 to 2023 in a field trial utilizing deficit irrigation.
- Key Findings:
- A total of 131 significant markers were identified as being associated with biomass yield, with 80 markers specifically linked to yield under drought stress. Notably, 19 of these markers were associated with multiple harvests.
- The study employed Genomic Best Linear Unbiased Prediction (GBLUP) to calculate predictive accuracies (PA) and genomic estimated breeding values (GEBVs).
- By removing low-informative SNPs (those with p-values > 0.05 from the additive GWAS model), the predictive accuracy increased by 47.3%.
- Conclusion: The identification of a high number of markers associated with yield under drought stress, along with the highest predictive accuracy of 0.9 achieved using GBLUP, marks a significant advancement in understanding and improving alfalfa yield in drought conditions.
I have some minor suggestions for the authors:
Reviewer Comments:
1- In abstract “and the highest PA (0.9) obtained using GBLUP in this study represent”, I think author missed full stop or some comma, in this sentence. Please restructure this sentence.
2- Line 217, 218, 229: please write the full name of genes when they appear first time in the text and try to avoid using several abbreviations.
3- Line 282, 285, 287, etc.: So many abbreviations, will make the manuscript different to read and understand for students and readers.
4- Section 4,.4. line 398: how much was the plant sample weighted for DNA extraction? Which plant tissue was used?
5- Line 408: why the authors aligned the Cleaned files with the Medicago sativa cultivar? Why not a variety used for alignment? Cultivar may have some variations….
Author Response
## Reviewer 3
- In abstract “and the highest PA (0.9) obtained using GBLUP in this study represent”, I think author missed full stop or some comma, in this sentence. Please restructure this sentence.
Done. The sentence was rephrased to “The high number of markers associated with yield under drought stress, and the highest PA (0.9) represents a significant achievement in improving yield under drought stress in alfalfa.”
- Line 217, 218, 229: please write the full name of genes when they appear first time in the text and try to avoid using several abbreviations.
Done, the full name of genes have been included in this version of the manuscript (L229-234, and L242-250).
- Line 282, 285, 287, etc.: So many abbreviations, will make the manuscript different to read and understand for students and readers.
Thank you for the comment, we removed the unnecessary abbreviations (L303)
- Section 4,.4. line 398: how much was the plant sample weighted for DNA extraction? Which plant tissue was used?
Genomic representations of 436 individual plants (one plant per family) were generated by collecting two to three leaflets (~100 mg) per genotype for DNA extraction and genotyping. We included this information in L428-430.
- Line 408: why the authors aligned the Cleaned files with the Medicago sativa cultivar? Why not a variety used for alignment? Cultivar may have some variations….
Thank you for your comment. In the alignment process, several genomes of Medicago sativa are available. However, we chose the XinJiangDaYe monoploid genome because it is utilized by Breeding Insight with the DArTag 3K array. Our goal was to ensure that the coordinates of marker positions align with those used in other studies for consistency and comparability.
Round 2
Reviewer 1 Report
Comments and Suggestions for Authors
Authors addressed all the reviewer's comment